# Development of a Machine Learning Model to Discriminate Mild Cognitive Impairment Subjects from Normal Controls in Community Screening

**DOI:** 10.3390/brainsci12091149

**Published:** 2022-08-28

**Authors:** Juanjuan Jiang, Jieming Zhang, Chenyang Li, Zhihua Yu, Zhuangzhi Yan, Jiehui Jiang

**Affiliations:** 1School of Communication and Information Engineering, Shanghai University, Shanghai 200444, China; 2Institute of Biomedical Engineering, School of Life Science, Shanghai University, Shanghai 200444, China; 3Shanghai Geriatric Institute of Chinese Medicine, Shanghai University of Traditional Chinese Medicine, Shanghai 200031, China

**Keywords:** mild cognitive impairment, neuropsychological tests battery, machine learning, screening tool

## Abstract

**Background**: Mild cognitive impairment (MCI) is a transitional stage between normal aging and probable Alzheimer’s disease. It is of great value to screen for MCI in the community. A novel machine learning (ML) model is composed of electroencephalography (EEG), eye tracking (ET), and neuropsychological assessments. This study has been proposed to identify MCI subjects from normal controls (NC). **Methods**: Two cohorts were used in this study. Cohort 1 as the training and validation group, includes184 MCI patients and 152 NC subjects. Cohort 2 as an independent test group, includes 44 MCI and 48 NC individuals. EEG, ET, Neuropsychological Tests Battery (NTB), and clinical variables with age, gender, educational level, MoCA-B, and ACE-R were selected for all subjects. Receiver operating characteristic (ROC) curves were adopted to evaluate the capabilities of this tool to classify MCI from NC. The clinical model, the EEG and ET model, and the neuropsychological model were compared. **Results**: We found that the classification accuracy of the proposed model achieved 84.5 ± 4.43% and 88.8 ± 3.59% in Cohort 1 and Cohort 2, respectively. The area under curve (AUC) of the proposed tool achieved 0.941 (0.893–0.982) in Cohort 1 and 0.966 (0.921–0.988) in Cohort 2, respectively. **Conclusions**: The proposed model incorporation of EEG, ET, and neuropsychological assessments yielded excellent classification performances, suggesting its potential for future application in cognitive decline prediction.

## 1. Introduction

Alzheimer’s disease (AD) is the most common neurodegenerative brain disease that affects 50–70% of patients with cognitive impairments over the age of 65 [1]. AD pathology leads to an irreversible deterioration in cognitive functions such as loss of memory, executive dysfunction, and attention disorders [2,3,4]. Mild cognitive impairment (MCI) refers to the intermediate period between the typical cognitive decline of normal aging and the more severe decline associated with dementia (e.g., AD) [5,6,7]. Because of the irreversibility of AD, it is of great value to screen MCI subjects at the community level [5,8,9].

Currently, biochemical tests (e.g., Cerebrospinal Fluid and Blood) and neuroimaging tests (e.g., Magnetic Resonance Imaging,) were considered efficient screening tools for MCI [10,11,12]. However, these techniques were usually invasive and expensive, restricting large-scale screening applications in the community [13,14]. Therefore, an effective and low-cost detectable approach to cognitive decline in MCI is urgently required.

Recently, MCI screening has attracted immersive interests. A Neuropsychological Tests Battery (NTB) is well recognized in the diagnostic pipelines of preclinical AD [15]. Multiple preclinical neuropsychological measures significantly predicted progression to AD from MCI and detected changes in patients in verbal and visual memory, visuospatial processing, error control, and subjective neuropsychological complaints [16]. Paul et. al. confirmed that neuropsychological tests quick-MCI to assess cognitive status in 3–5 min and can discriminate MCI accurately in primary care [17]. Neuropsychological tests were clearly appropriate for MCI community screening, as are emerging cognitive assessments such as electroencephalogram (EEG) and eye tracking (ET) to monitor cognitive function. Murty et al. found that stimulus-induced gamma rhythms from EEG were significantly lower in MCI/AD subjects compared to their age- and gender-matched controls, suggesting that gamma of EEG could be used as a potential screening tool for MCI or AD in humans [18]. Oyama et al. developed a brief cognitive assessment utilizing an eye-tracking technology that can enable quantitative scoring and the sensitive detection of cognitive impairment in patients with mild cognitive impairment and dementia [19]. Nie et al. found that eye movement parameters are stable indicators to distinguish patients with MCI and cognitively normal subjects and are not affected by different testing versions and numbers [20]. The incorporation of neuropsychological tests and physiological measurements warrants further study as a practical and cost-effective method for wide-scale screening for identifying older adults who may be at risk for pathological cognitive decline. Neuropsychological tests might be limited in their effectiveness in MCI screening while acknowledging that neuropsychological tests are inadequate for making a definitive diagnosis. To increase the precision and sensitivity of MCI screening, several researchers incorporated NTB into objective physiological measures, such as prefrontal EEG [21] and ET [22]. For instance, our previous work validated the feasibility of physiological measures using EEG and ET in distinguishing MCI from HC, with a classification accuracy of 81.5% [23].

In addition, with the development of artificial intelligence techniques, machine learning (ML) methods have been widely used for the differential diagnosis of MCI [15,23,24,25]. For example, Lin et al. developed non-invasive clinical variables and ML classifiers, including Support Vector Machine (SVM), Logistic Regression (LR), and Random Forest (RF), to achieve over 75% classification accuracy to classify subjects who converted to MCI from normal within four years [25]. Yim et al. proposed a ML algorithm to identify cognitive dysfunction based on neuropsychological tests including the Montreal Cognitive Assessment (MoCA). The results showed a good classification performance between cognitive impairment and normal subjects [15]. However, there were few models using neuropsychological tests, physiological tests, and ML algorithms in the previous studies.

This study aims to propose and validate a novel and low-cost screening model consisting of neuropsychological tests, physiological tests, and ML algorithms. Importantly, to evaluate the robustness of the model, two independent cohorts were used in this study.

## 2. MCI Prediction Algorithms

Figure 1 shows the flowchart of the proposed model, which was composed of four steps: data collection, data preprocessing, feature extraction and selection, and classification based on ML classifiers. These steps were described in detail as the following:

### 2.1. Data Collection

EEG, ET, neuropsychological test (Appendix A), and demographic data (age, gender, and education) were selected as the inputs of the model. Details of the data collection step were described in our previous study [23] and provided in the Appendix A. 

### 2.2. Data Preprocessing

This model included an automatic data preprocessing step for EEG, ET, and NTB.

#### 2.2.1. EEG Preprocessing

Invalid EEG data was first removed according to whether the EEG electrode was offset. Next, the power frequency noise, electromyogram signal, electrocardiogram signal, and other external noises were removed using a band stop filter and a band pass filter. Simple second-order Butterworth filtering was applied with a passband of 0.5–30 Hz. Finally, we overlapped 60% of the EEG data by applying a 5 s moving window, providing 15 overlapping segments for each subject. The EEG signal was preprocessing using EEGLAB toolbox implemented in MATLAB 2018a (Math Works Inc., Sherborn, MA, USA).

#### 2.2.2. ET Preprocessing

First, excessive noise from ET data was eliminated. Next, the gaze position signal was normalized to the display coordinates to avoid the interpolation bias. Finally, a low pass Butterworth filter with a cut-off frequency of 5 Hz was implemented in MATLAB 2018a (Math Works Inc., Sherborn, MA, USA).

#### 2.2.3. NTB Data Preprocessing

NTB data were cleaned, and all abnormal values were eliminated. Finally, neuropsychological test scores were normalized into 0–1.

### 2.3. Feature Extraction

#### 2.3.1. EEG Data

Frequency-domain and spectral-domain features of the EEG signal were extracted. A Fourier transform of the autocorrelation function was employed to transform the EEG signal from time-domain to frequency-domain to get the power spectral density. Four EEG frequency bands (delta 0.5–4 Hz, theta 4–8 Hz, alpha 8–13 Hz, and beta 13–30 Hz) were filtered in this study. The power spectrum of each frequency band and specific spectral power ratios like the alpha/theta power ratio was computed. The extracted linear features of the EEG were consistent with our preliminary work [23]. Nonlinear features of the EEG, including approximate entropy (ApEn) [26], Multiscale entropy (MsEn) [27], and Lempel Ziv complexity (LZC) were calculated [28]. The calculation formulas of the EEG features were described in the section of Feature extraction and selection of Appendix A.

#### 2.3.2. ET Data

ET data was divided into saccade data and gaze data. The association of gazes and saccades with specific regions on visual stimuli was examined. Then, visual scan parameters such as blink frequency, blink time, fixation time, and sustained attention duration were calculated. The nonlinear features of ET were extracted by LZC.

#### 2.3.3. NTB Data

NTB data, which are numerical, included subtest scores, total test scores, and response time. Meaningful numerical features were subsequently converted to z-scores using Z transformation.

### 2.4. Feature Selection

The Minimum Redundancy-Maximum Relevance (MRMR) algorithm was used for feature selection [29]. In the MRMR algorithm, the correlation between different feature subsets is modeled as:(1)$$\Theta =\frac{1}{\left|\Omega \right|}\sum_{m}\sum_{f_{i}\in \Omega }M\left(f_{i},m\right)$$
where the feature subset Ω is from the feature set F and $F=\left\{f_{1},\ldots ,f_{D}\right\}$. In this tool, m={+1,−1} represents HC and MCI respectively and M is the mutual information between the feature subset and the target classes which is given by
(2)$$M\left(X,Y\right)=\sum_{X}\sum_{Y}p\left(X,Y\right)\log_{2}\left(\frac{p\left(X,Y\right)}{p\left(X\right)p\left(Y\right)}\right)$$
where p(X), p(Y), p(X,Y) are the marginal probability distributions and joint probability distributions of variable X, Y respectively. Clearly, the mutual information comes to zero when p(X,Y)=p(X)p(Y), which states that the feature is independent with the target classes.

The redundancy between the feature $f_{i}$ and other features can be modeled as:(3)$$\Delta_{\Omega ,f_{i}}=\frac{1}{|\Omega {|}^{2}}\sum_{f_{j}\subset \Omega ,f_{i}\neq f_{j}}M\left(f_{i},f_{j}\right)$$

Thus, the feature meeting the minimum redundancy-maximum correlation principle can be obtained via:(4)$$f_{i}^{*}=\underset{f_{i}\subset \Omega }{\arg\max}\frac{\Theta }{\Delta_{\Omega ,f_{i}}}$$

In the above equation, the optimal features can be obtained by maximizing the correlation between the features and the target classification and minimizing the redundancy between the features. By performing similar operations on different feature subsets, multiple optimal features can be found to reduce the complexity and improve the algorithm decision performance.

### 2.5. Classification

A support vector machine (SVM) was used as the ML classifier with Anaconda Spyder 3.7 (Anaconda Inc., Austin, TX, USA). As a classic supervised learning method, SVM has been widely used in statistical classification and regression analysis due to its ability to map vectors linearly to a higher dimensional space that creates a maximum margin hyperplane to achieve high classification performance.
(5)$$w^{T}x+b=0$$

Support vectors maximize the margin of the classifier by changing the position and orientation of the hyperplane. Kernel functions of SVM or “kernel trick” by SVM were applied to remedy the issue that the points are not separable linearly due to the position of the data. Kernel trick involves the transformation of the existing algorithm from a lower-dimensional data set to a higher one. The amount of information remains the same, but in this higher dimensional space, it is possible to create a linear classifier. Several K kernels are assigned to each point which then helps determine the best fit hyperplane for the newly transformed feature space. With enough K functions, it is possible to get precise separation.

Linear SVM classifier with hard margin:(6)$$W(\alpha )=-\sum_{i=1}^{l}\alpha_{i}+\frac{1}{2}\sum_{i=1}^{l}\sum_{j=1}^{l}y_{i}y_{j}\alpha_{i}\alpha_{j}X_{i}X_{j}$$

Kernel trick equation minimizing W subject to:(7)$$\sum_{i=1}^{l}y_{i}\alpha_{i}=0\;0\leq \alpha_{i}\leq C$$

## 3. Materials and Methods

### 3.1. Subjects

We recruited two cohorts for this study. Cohort 1 was composed of 336 subjects from four communities in Jiading district, Shanghai, China, including 152 MCI patients and 184 normal controls (NC) subjects. Cohort 2 was composed of 44 MCI patients and 48 NC subjects from one community in Baoshan district, Shanghai, China. All subjects also underwent a battery of cognitive evaluations, including Addenbrooke’s Cognitive Examination-revised (ACE-R) and Montreal cognitive assessment-basic (MoCA-B). The permission of MoCA-B in the study was received via https://www.mocatest.org/permission (accessed on 28 June 2017).

All subjects signed an informed consent before the examinations. This study has been approved by the ethics committee of Long Hua Hospital in Shanghai University of Traditional Chinese Medicine (Ethical number: 2017LCSY345) and conducted in accordance with the principles of the Declaration of Helsinki. In this study, Cohort 1 was used as the training and validation group to train the SVM classifier. Cohort 2 was used as an independent test group to verify the robustness of the classification results.

MCI was defined by an actuarial neuropsychological strategy proposed by Jak and Bondi [30], subjects were considered to have MCI if they met any of the following three criteria and neglected to meet the criteria for dementia. The inclusion criteria for MCI were as follows [31,32]: (1) right-handed, and Mandarin-speaking subjects; (2) a subjective memory complaint; (3) memory impairment relative to age and education-matched healthy elderly individuals confirmed by performance on neuropsychological assessments (below 1.5 standard deviations); (4) intact general cognitive function confirmed by MoCA-B scores ≥ 26; (5) intact activities of daily living; and (6) without dementia confirmed by a physician.

Exclusion criteria of MCI were as follows: (1) other neurological diseases including cerebrovascular disease, brain trauma, Parkinson’s syndrome, brain tumor, and epilepsy; (2) current major psychiatric disease such as severe depression and anxiety; (3) other neurological conditions that could cause cognitive decline (e.g., brain tumors, Parkinson’s disease, encephalitis, or epilepsy) rather than AD spectrum disorders; (4) systemic diseases that may lead to cognitive decline (thyroid dysfunction, severe anemia, syphilis, or HIV, etc.); (5) other conditions such as a history of CO poisoning and general anesthesia; (6) severe visual or hearing impairment; (7) contraindication for MRI.

The inclusion criteria for NC included the following: (1) no subjective or informant-reported memory decline; (2) non-clinical depression (Geriatric Depression Scale scores < 6); (3) normal age-adjusted, gender-adjusted, and education-adjusted performance on standardized cognitive tests.

### 3.2. Data Acquisition

All data were selected from 1 September 2017 to 31 August 2018 in the communities, Shanghai, China. The data selection protocol has been introduced in the Appendix A.

### 3.3. Validation Experiments for Optimal Parameters of the Classifier

We adjusted the hyper-parameters for the SVM classifier such as kernel function, penalty factor C, and coefficient of kernel function gamma with good classification performance by 5-fold cross-validation. Different kernels, including linear, polynomial, and RBF were compared in this study. Cohort 1 was used to train these parameters.

### 3.4. Discriminative Analysis

The classification results from four models were compared by using the SVM classifier, including (1) the clinical model (clinical variables including age, gender, educational level, MoCA-B, ACE-R), (2) the single neuropsychological test model (20 subtests of NTB showed in the Appendix A), (3) the single physiological test model (EEG and ET), and (4) the proposed tool model. We used the 5-fold cross-validation method to calculate the classification results.

### 3.5. Statistical Analysis

Differences in demographic and cognitive performance between the NC group and the MCI group were evaluated by two sample *t*-tests or chi-square (χ^2^) tests of Statistical Package V24 for Social Sciences (SPSS Inc., Chicago, IL, USA). The significance level was set as *p* < 0.05. Receiver operating characteristic (ROC) curves were used to evaluate the capabilities of the tool in distinguishing MCI from NC. The areas under the curves (AUCs) with 95% confidence intervals (CIs) were calculated.

## 4. Results

### 4.1. Demographic and Clinical Characteristics

The detailed demographic and clinical characteristics were reported in Table 1. The results showed that the scores of MoCA-B and ACE-R from MCI patients were significantly lower than NC’s scores (*p* < 0.001, two-sample *t*-test). There were no significant differences in age (*p* = 0.875; two-sample *t*-test), gender (*p* = 0.541; chi-square test) or years of education (*p* = 0.071; Wilcoxon rank-sum test) of cohort 1. There were no significant differences in age (*p* = 0.783; two-sample *t*-test), gender (*p* = 0.492; chi-square test) or years of education (*p* = 0.068; Wilcoxon rank-sum test) of cohort 2 either.

### 4.2. Validation Experiments for Optimal Parameters of Classifier

The best classification performance was obtained under the specific parameters (C = 1.1, GAMMA = 0.001) while the kernel function was set to RBF. Table 2 shows the detailed performance of different kernel functions and corresponding parameters.

### 4.3. Discriminative Analysis

Table 3 and Table 4 showed comparison results of four models in Cohort 1 and 2, respectively. Classification results showed that the performance of the proposed tool was better than other models (Accuracy: 84.5 ± 4.43%; Sensitivity: 81.9 ± 7.88%; Specificity: 86.8 ± 6.19%; AUC: 0.942 (0.893–0.982)) in Cohort 1. Classification results also showed that the performance of the proposed tool was better than other models (Accuracy: 88.8 ± 3.59%; Sensitivity: 86.2 ± 6.46%; Specificity: 91.0 ± 5.39%; AUC: 0.966 (0.921–0.988)) in Cohort 2. Figure 2 and Figure 3 showed the ROC results of the four models in both cohorts.

## 5. Discussion

Cognitive decline remains highly underdiagnosed in the community despite extensive efforts to find novel approaches to detect MCI and find objective screening methods for cognitive decline could improve early MCI diagnosis. MCI screening in the community has become a hot topic nowadays. In light of their excellent performance in detecting a cognitive decline in MCI patients, multimodal detection approaches have been commonly used in computer-aided disease diagnostic fields of community screening. In this study, we proposed a ML model based on EEG, eye movement, and neuropsychological tests for MCI screening at the community level. In contrast to other traditional models, such as the EEG-based model, ET-based model, and NTB-based model, the classification results of our model outperformed other traditional models.

So far, a lot of studies have focused on the classification of NC and MCI by using machine learning models for screening in primary care. For instance, Siuly et al. performed a Piecewise Aggregate Approximation (PAA) technique for compressing massive volumes of EEG data for reliable analysis and developed a model based on Extreme Learning Machine (ELM) with permutation entropy (PE) and auto-regressive (AR) model features to achieve the highest MCI classification accuracy (98.8%) [33]; Lagun et al. applied a SVM based machine learning model to reach the accuracy of 87% to detect MCI by modeling eye movement characteristics such as fixations, saccades, and refixations during the Visual Paired Comparison (VPC) task [34]; Yim et al. developed a screening model based on a gradient boosting (GB) algorithm to identify MCI by neuropsychological test results and reached the classification accuracy of 93.5% [15]; and, Wang et al. developed a Random Forest (RF)-based model to optimize the content of cognitive evaluation and achieved an accuracy of 68% in the classification of MCI and NC [35].

Notably, our classification results were similar to previous studies, indicating the reliability of our results. As shown in Table 5, although previous studies based on EEG analysis performed powerful discrimination for MCI detection (ACC = 98.8% in Siuly’s model), it is worth noting that these studies based on expensive and long-term physiological signal collection devices are seldom used in primary care. By contrast, the wearable EEG device used in our approach was more suitable for large-scale MCI screening. In contrast to earlier studies based on ET and NTB, our method achieved better accuracy. Additionally, the advantages of our method were also summarized as follows:

(1)In terms of feature extraction, the linear and nonlinear feature analysis has been successfully used to identify the powerful biomarkers of neurophysiological diseases, such as Alzheimer’s disease (AD). In this study, we applied both linear and nonlinear methods to extract EEG and eye movement features. For EEG, complexity analysis as a nonlinear dynamic method can represent the rate of new patterns appearing in a time series, and to a certain extent, details of the signal can be presented in the binarized sequence.(2)In terms of feature selection and classification, the SVM model was selected. As a ML model, the SVM is suitable for classifying the features obtained from neuropsychological assessments.(3)In terms of the clinical setting, we depicted a machine learning framework for automated cognitive assessment data analysis for the precise classification of healthy and mild cognitive impairment individuals. Our work opens the possibility for automated assessment of cognitive function in community screening.

Although our proposed method achieved a good classification of screening MCI and NC, several limitations still exist. First, the whole experiment is time-consuming and thus leads to a decrease in the degree of completion and cooperation of patients. Second, the de-noising algorithm may influence the results of feature extraction and classification. Third, the sample size of NC and MCI individuals was limited, and increasing the sample size in future studies should be taken into consideration. Longitudinal imaging studies are still absent. In the subsequent research, ongoing follow-up observational studies of individuals will facilitate the investigation and validation of our results. Finally, SVM was only used as the classifier in this study. If alternative classifiers such as using extreme learning machines or deep learning models were developed, better classification results will be obtained.

## 6. Conclusions

In this study, an automatic and non-invasive MCI detection model was proposed, which integrated EEG, Eye movement techniques, and a neuropsychological test battery. The results indicated the potential application for MCI detection and guided referral for a more comprehensive evaluation to ultimately facilitate early intervention in primary care.

## Figures and Tables

**Figure 1 brainsci-12-01149-f001:**
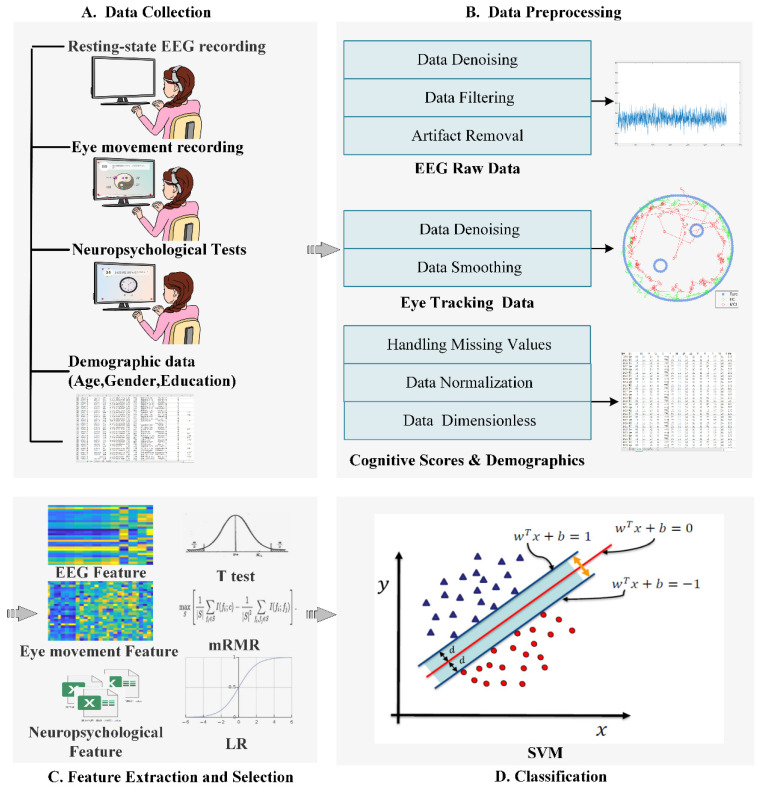
The flowchart of the proposed model.

**Figure 2 brainsci-12-01149-f002:**
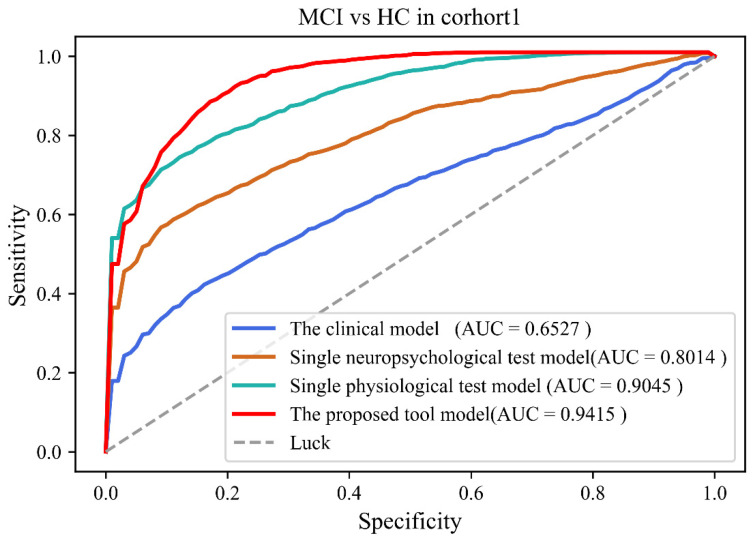
The receiver operating curves of four models in cohort 1.

**Figure 3 brainsci-12-01149-f003:**
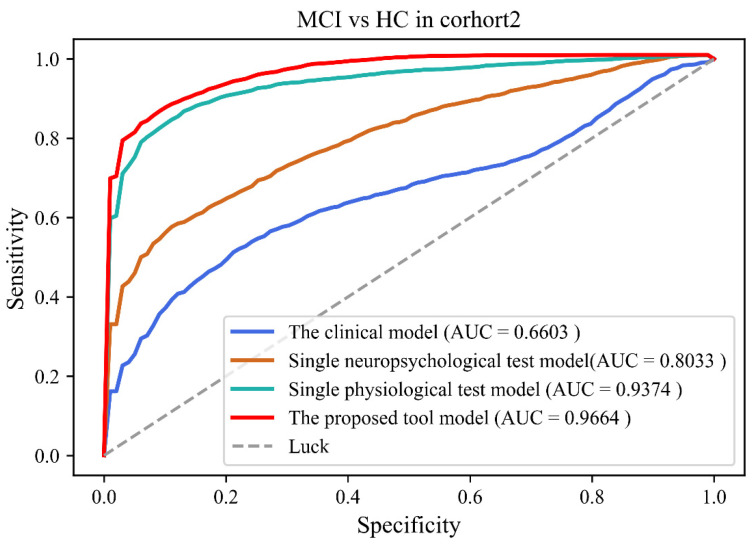
The receiver operating curves of four models in cohort 2.

**Table 1 brainsci-12-01149-t001:** Demographic and clinical characteristics of subjects.

	Cohort 1			Cohort 2		
	NC (184)	MCI (152)	*p* Value	NC (48)	MCI (44)	*p* Value
Age (years)	71.7 ± 4.66	71.6 ± 4.15	0.875 ^b^	69.3 ± 17.0	76.5 ± 11.3	0.783 ^b^
Education (years)	9.36 ± 3.47	8.16 ± 3.74	0.541 ^a^	13.0 ± 3.87	10.8 ± 5.66	0.492 ^a^
Gender (male/female)	101/83	78/74	0.071 ^c^	18/30	16/28	0.068 ^c^
MoCA-B	28.3 ± 0.95	23.2 ± 3.40	<0.001 ^b^ *	23.8 ± 3.28	16.4 ± 3.84	<0.001 ^b^ *
ACE-R	72.1 ± 7.79	63.7 ± 8.53	<0.001 ^b^ *	71.0 ± 24.9	64.2 ± 8.28	<0.001 ^b^ *

Note: Data are presented as mean ± standard deviation. * Indicates a statistical difference between groups, *p* < 0.05; ^a^: the *p* value was obtained by χ2 test, ^b^: the *p* value was obtained by two-sample *t* tests, ^c^: the *p* value was obtained by Wilcoxon rank-sum test. Abbreviations: NC, normal control; MCI, Mild Cognitive Impairment; MoCA-B, Montreal cognitive assessment-basic; ACE-R, Addenbrooke’s Cognitive Examination Revised.

**Table 2 brainsci-12-01149-t002:** The optimized hyper-parameters of SVM in test dataset.

Kernel Function	C	GAMMA	Accuracy (%)	Sensitivity (%)	Specificity (%)	AUC (95% CI)
Linear	4.0	/	84.3 ± 4.05	83.1 ± 7.70	85.5 ± 4.97	0.906 (0.841–0.969)
Poly	20.0	0.02	78.1 ± 9.90	83.6 ± 10.6	71.3 ± 13.6	0.851 (0.747–0.954)
**RBF**	**1.1**	**0.001**	**84.5 ± 4.34**	**82.4 ± 7.36**	**86.5 ± 6.51**	**0.934 (0.878–0.977)**
Sigmoid	17.0	0.01	82.1 ± 6.08	90.9 ± 8.13	71.3 ± 11.7	0.851 (0.838–0.964)

C represents the regularization coefficient, gamma represents the kernel function coefficient, AUC represents the area under the ROC curve, the bold part in the table is the optimal value of each column, and the values in the table are the mean and standard deviation after five cross-validations.

**Table 3 brainsci-12-01149-t003:** The classification results of four models in cohort 1.

Comparative Model	Accuracy (%)	Sensitivity (%)	Specificity (%)	AUC (95% CI)
The clinical model	62.6 ± 5.19	54.7 ± 6.81	71.4 ± 5.56	0.653 (0.541–0.783)
Single neuropsychological test model	75.6 ± 4.60	55.7 ± 8.15	71.2 ± 4.72	0.8014 (0.700–0.885)
Single physiological test model	81.4 ± 4.66	72.1 ± 8.25	89.2 ± 5.42	0.9045 (0.819–0.961)
The proposed tool model	84.5 ± 4.43	81.9± 7.88	86.8 ± 6.19	0.9415 (0.893–0.982)

**Table 4 brainsci-12-01149-t004:** The classification results of four models in cohort 2.

Comparative Model	Accuracy (%)	Sensitivity (%)	Specificity (%)	AUC (95% CI)
The clinical model	65.7 ± 4.93	43.3 ± 10.6	90.1 ± 7.94	0.660 (0.543–0.789)
Single neuropsychological test model	75.0 ± 5.22	54.1 ± 8.63	91.5 ± 4.73	0.803 (0.681–0.889)
Single physiological test model	87.0 ± 4.27	82.4 ± 7.94	90.6 ± 5.05	0.937 (0.867–0.985)
The proposed tool model	88.8 ± 3.59	86.2 ± 6.46	91.0 ± 5.39	0.966 (0.921–0.988)

**Table 5 brainsci-12-01149-t005:** The performance of analogous MCI detection methods in the literature.

Detection Tools	Modality	Subject	Method	Classifier	Accuracy
EEG based	Siuly, 2020 [33] EEG (19 Electrodes)	27	EEG features	ELM	98.8%
ET based	Lagun, 2011 [34] ET Test	174	ET features	SVM	87%
Neuropsychological test based	Yim, 2020 [15]	614	The mean total scores of neuropsychological test	GB	93.5%
NTB based	Wang, 2022 [35] Neuropsychological tests battery	241	NTB scores	RF	68%
Proposed Method NTB, EEG and Eye tracking	EEG (1 electrode) & ET & Neuropsychological test battery	336	EEG & ET features & NTB scores	SVM	88.8%

## Data Availability

The data that support the findings of this study are available from the corresponding author.

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
