# Peer review of "Development of a Machine Learning Model to Discriminate Mild Cognitive Impairment Subjects from Normal Controls in Community Screening"

_brainsci, 2022, doi:10.3390/brainsci12091149_

Round 1
Reviewer 1 Report
In this manuscript, the authors claimed that they have introduced a new “tool” for MCI-HC discrimination. However, what the manuscript presents is in fact a classifier with an added GUI.
Strength:
1. Considerably large sample size
2. High accuracy that is adequately tested on separate and additional sample
Weakness:
1. The manuscript writing requires a thorough proofread. There are several instances that the authors’ pros are not clear. A few examples include:
a. Line 34: “most common neurodegenerative brain disease.” I am not quite sure about this statistics. It is important for the authors to cite their source for readers’ future reference and better understanding.
b. Line 46: “Neuropsychological tests combing physiological tests could be the alternative tool.”: This line is not very clear. Do the authors mean combination of such tests?
c. Line 51: “The weakness of traditional neuropsychological tests is the subjectivity.” This is not clear. Do you mean such tests do not consider the subjective responses of the individuals into consideration? If so, the authors’ claim is too strong since these tests are in fact designed to capture the individuals’ subjective responses (although might be limited in their effectiveness).
d. Line 73: “Figure 1 showed ” to “Figure 1 shows”
Additionally, there are typos, for instance:
a. Lines 46 and 68: “combing” to “combining” (also the heading of Section 2 .. there could be more. Please check for other potential instances).
b. In Fig1. “Artifact Renovating” to “Artifact Removal” and “Handing missing values” to “Handling missing values” and “Data Normalizaing” to “Data Normalization”
Also, please be consistent in capitalizing words (either capitalize or not).
The authors’ results are quite impressive. However, I am not quite sure if the content of this manuscript can be passed as a “new tool.”
In case of the latter, the authors are required to provide a more comprehensive literature review on the state-of-the-arts on such classification studies and also highlight how their study improve + help further progress in this field.
Author Response
Dear Editors and Reviewers,
Thank you for your letter and for the reviewers’ comments concerning our manuscript entitled “A novel tool to discriminate mild cognitive impairment subjects from normal controls in the community screening”. Those comments are all valuable and very helpful for revising and improving our paper, as well as the important guiding significance to our researches. We have studied these comments carefully and have made corrections which we hope meet with approval. The main corrections in the paper and the response to the reviewer 1’s comments are as follows. All revisions were with color red.

Reviewer 2 Report
In the manuscript entitled “A novel tool to discriminate mild cognitive impairment subjects from normal controls in the community screening”, Juanjuan Jiang and colleagues showed that a novel screening tool derived from electroencephalography (EEG), eye tracking (ET) and neuropsychological assessments to diagnosis mild cognitive impairment (MCI) patients in the clinical screening. The classification results showed that this tool could yield excellent classification performances, suggesting its potential for future community application. However, I cannot recommend publication in the current form due to the following additional issues which should be addressed.
Q1: in Abstract, the author developed three classification models (clinical, EEG and ET, neuropsychological models) and compared these classification results. It would be better if you combined all information and further performed your experiment. Because the clinical and neuropsychological information are available when collecting EEG and ET signals.
Q2: in Introduction Line 51, ‘The weakness of traditional neuropsychological tests is the subjectivity.’ Please add more detailed description for this sentence.
Q3: Because the study involved human experimentation, approval by the Ethics Review Committee and informed consent from participants are required. Please supplement section 3.1 with the relevant ethical statement to meet the Helsinki Ethical Requirements.
Author Response
Dear Editors and Reviewers,
Thank you for your letter and for the reviewers’ comments concerning our manuscript entitled “A novel tool to discriminate mild cognitive impairment subjects from normal controls in the community screening”. Those comments are all valuable and very helpful for revising and improving our paper, as well as the important guiding significance to our researches. We have studied these comments carefully and have made corrections which we hope meet with approval. The main corrections in the paper and the response to the reviewer 2’s comments are as follows. All revisions were with color red.
In the manuscript entitled “A novel tool to discriminate mild cognitive impairment subjects from normal controls in the community screening”, Juanjuan Jiang and colleagues showed that a novel screening tool derived from electroencephalography (EEG), eye tracking (ET) and neuropsychological assessments to diagnosis mild cognitive impairment (MCI) patients in the clinical screening. The classification results showed that this tool could yield excellent classification performances, suggesting its potential for future community application. However, I cannot recommend publication in the current form due to the following additional issues which should be addressed.
Point 1: in Abstract, the author developed three classification models (clinical, EEG and ET, neuropsychological models) and compared these classification results. It would be better if you combined all information and further performed your experiment. Because the clinical and neuropsychological information are available when collecting EEG and ET signals.
Response 1: Thank you for your comment. Actually, our proposed model, the integrated neuro-psychological and physiological test model, which had included clinical information. Therefore, this model had employed all information obtained from data collection and could be regarded as a combined model. According to your suggestion, we have rewritten the sentence in Abstract section in Line 21 to Line 22.
Changes in the manuscript:
EEG, ET, Neuropsychological Tests Battery (NTB) and clinical variables with age, gender, educational level, MoCA-B, ACE-R were selected for all subjects.
Point 2: in Introduction Line 51, ‘The weakness of traditional neuropsychological tests is the subjectivity.’ Please add more detailed description for this sentence.
Response 2: Thank you for your comment. According to your suggestion, we have rewritten the sentence and updated in the manuscript. Please check this change in Line 67 to 69.
Changes in the manuscript:
Neuropsychological tests might be limited in their effectiveness in MCI screening, while acknowledging that neuropsychological tests inadequate for making a definitive diagnosis.
Point 3: Because the study involved human experimentation, approval by the Ethics Review Committee and informed consent from participants are required. Please supplement section 3.1 with the relevant ethical statement to meet the Helsinki Ethical Requirements.
Response 3: Thank you for your comment. According to your suggestion, we have supplied the relevant ethical statement and rewritten the sentence. Please check these ethical statements in line 183 to 186.
Changes in the manuscript:
All subjects signed an informed consent before the examinations. This study has been approved by the ethics committee of Long Hua Hospital in Shanghai University of Traditional Chinese Medicine (Ethical number: 2017LCSY345) and conducted in accordance with the principles of the Declaration of Helsinki.

Reviewer 3 Report
The authors developed this study with the aim of presenting a new methodology to detect Mild Cognitive Impairment (MCI) and discriminate it from healthy subjects. To do this, they first conducted a brief presentation of the definitions of Alzheimer Disease (AD) and MCI and the tools used in the literature for their diagnosis, affirming the need to identify an alternative and less costly method. This is an interesting idea in light of the need to identify cognitive decline as soon as possible and take prompt action to slow down the onset of AD.
However, I ask the authors to consider the following comments to improve and implement the quality and robustness of the manuscript.
Comment 1: The quality of English needs to be improved, particularly in the use of some terms. Authors are encouraged to review the manuscript. The use of appropriate terminology is critical to the context. Authors are encouraged to rephrase some sentences and review the use of some terms. Here are some references: line 46 and 68, the term "combing" does not concern the scientific field, we invite you to review the meaning of the sentence and modify the term used. Line 118-119 the significance inherent in how the data from the neuropsychological assessment was handled is unclear. Line 261, please correct “compare”. Furthermore, given the complexity of the research project and the objectives presented, in my opinion the use of the term "tool", which usually refers to the use of a single "instrument", could be misleading and reductive in the presentation of the project that instead proposes a complex methodology that includes the use of different tools (EEG, ET, ML and neuropsychological assessment). Authors are therefore invited to choose a more appropriate term and use it throughout the paper.
Comment 2: The introduction is interesting to present the topic of the need to propose a new methodology to detect MCI. However, it is believed that the bibliography used to present the information currently present in the literature is not sufficient. In particular, it is necessary to explain and support with bibliographic references the statement in line 51 "the weakness of traditional neuropsychological tests is the subjectivity". In fact, the process that allows to build an appropriate and correctly validated neuropsychological test usually leaves no room for subjectivity, as it is always calibrated and standardized tools. It is therefore considered necessary that the meaning of this statement be explained and contextualized with the use of an appropriate bibliography. In addition, in lines 59 and 62, two papers are cited without bibliographic references. Please implement these aspects and elaborate on these paragraphs, supporting your statements with appropriate bibliographic references.
Comment 3: in the paragraph Methods the inclusion and exclusion criteria should be better specified and clarified as they are vague and generic for the control group. It is important to explain how the authors detect no memory decline in these subjects. Did they use some neuropsychological or other assessment?
Comment 4: In the Discussion section it might be beneficial to discuss the results in more depth in the context of the clinical setting using adequate bibliography.
Comment 5: Authors are asked to pay attention to citations in the text. In the bibliography there are no bibliographic references of the neuropsychological tests that have been used and or cited in the text. Please enter the correct references.
Author Response
Dear Editors and Reviewers,
Thank you for your letter and for the reviewers’ comments concerning our manuscript entitled “A novel tool to discriminate mild cognitive impairment subjects from normal controls in the community screening” .Those comments are all valuable and very helpful for revising and improving our paper, as well as the important guiding significance to our researches. We have studied these comments carefully and have made corrections which we hope meet with approval. The main corrections in the paper and the response to the reviewer 3’s comments are as follows. All revisions were with color red.
The authors developed this study with the aim of presenting a new methodology to detect Mild Cognitive Impairment (MCI) and discriminate it from healthy subjects. To do this, they first conducted a brief presentation of the definitions of Alzheimer Disease (AD) and MCI and the tools used in the literature for their diagnosis, affirming the need to identify an alternative and less costly method. This is an interesting idea in light of the need to identify cognitive decline as soon as possible and take prompt action to slow down the onset of AD.
However, I ask the authors to consider the following comments to improve and implement the quality and robustness of the manuscript.
Comment 1: The quality of English needs to be improved, particularly in the use of some terms. Authors are encouraged to review the manuscript. The use of appropriate terminology is critical to the context. Authors are encouraged to rephrase some sentences and review the use of some terms.
Point 1: Here are some references: line 46 and 68, the term "combing" does not concern the scientific field, we invite you to review the meaning of the sentence and modify the term used.
Response 1: Thank you for your comment. The whole manuscript had double checked by native speaker. According to your suggestion, we have rewritten these sentences. Please check these changes in the revised manuscript.
Changes in the manuscript in Line 48-Line 49:
Recently, MCI screening has attracted immersive interests. Neuropsychological tests battery(NTB) were well recognized in the diagnostic pipelines of preclinical AD [15].
Changes in the manuscript in Line 84-Line 85:
This study aims to propose and validate a novel and low-cost screening model consisting of neuropsychological tests, physiological tests, and ML algorithms.
Point 2:Line 118-119 the significance inherent in how the data from the neuropsychological assessment was handled is unclear.
Response 2: Thank you for your comment. To facilitate statistical analysis and model construction, we performed Z-transformation on these neuropsychological data to fit a normal distribution. We have extended the description of feature extracted from Neuropsychological data in the Line 118-119. Please check these changes in Line 131 to 133.
Changes in the manuscript:
NTB data, which are numerical, included subtest scores, total test scores and response time. Meaningful numerical features were subsequently converted to z-scores using Z transformation.
Point 3: Line 261, please correct “compare”.
Response3: Thank you for your comment. We have replaced “Compare” to “In contrast to” in Line 261 and rewritten the following sentence in Line 299 to Line 300.
Changes in the manuscript:
In contrast to earlier studies based on ET and NTB, our method achieved better accuracy.
Point 4: Furthermore, given the complexity of the research project and the objectives presented, in my opinion the use of the term "tool", which usually refers to the use of a single "instrument", could be misleading and reductive in the presentation of the project that instead proposes a complex methodology that includes the use of different tools (EEG, ET, ML and neuropsychological assessment). Authors are therefore invited to choose a more appropriate term and use it throughout the paper.
Response4: Thank you for your comment. According to your suggestion, we have replaced “tool” to “machine learning model” throughout the manuscript.
Point 5: Comment 2: The introduction is interesting to present the topic of the need to propose a new methodology to detect MCI. However, it is believed that the bibliography used to present the information currently present in the literature is not sufficient.
Response 5: Thank you for your comment. According to your suggestion, we have added the description of new method for MCI screening from neuropsychological tests, EEG and eye movement modality transitioning from a single modality to MCI screening with multiple model fusion.
Changes in the manuscript:
Recently, MCI screening has attracted immersive interests. Neuropsychological tests battery(NTB) were well recognized in the diagnostic pipelines of preclinical AD [15]. Multiple preclinical neuropsychological measures significantly predicted progression to AD from MCI, and detected changes of patients in verbal and visual memory, visuospatial processing, error control, and subjective neuropsychological complaints [16]. Paul et. al. confirmed that neuropsychological test quick MCI to assess cognitive sta-tus in 3-5mins and discriminate MCI accurately in primary care [17]. Neuropsychological tests were clearly appropriate for MCI community screening, as are emerging cog-nitive assessments such as electroencephalogram (EEG) and eye tracking (ET) to mon-itor cognitive function. Murty et. al., found that stimulus-induced gamma rhythms from EEG were significantly lower in MCI/AD subjects compared to their age- and gender-matched controls, suggesting that gamma of EEG could be used as a potential screening tool for MCI or AD in humans[18].Oyama et. al., developed a brief cognitive assessment utilizing an eye-tracking technology which can enable quantitative scoring and the sensitive detection of cognitive impairment in patients with mild cognitive impairment and dementia [19]. Nie et al., found that eye movement parameters are stable indicators to distinguish patients with MCI and cognitively normal subjects and are not affected by different testing versions and numbers [20].
[17] Paúl, C.; Sousa, S.; Santos, P.; O’Caoimh, R.; Molloy, W.J.I.i.A. Screening Neurocognitive Disorders in Primary Care Services: The Quick Mild Cognitive Impairment Approach. Innovation in Aging 2020, 4, 158 - 158, doi:10.1093/geroni/igaa057.515.
[18] Murty, D.V.P.S.; Manikandan, K.; Kumar, W.S.; Ramesh, R.G.; Purokayastha, S.; Nagendra, B.; M L, A.; Balakrishnan, A.; Javali, M.; Rao, N.P.; et al. Stimulus-induced gamma rhythms are weaker in human elderly with mild cognitive impairment and Alzheimer’s disease. eLife 2021, 10,e61666, doi:10.7554/eLife.61666.
[19] Oyama, A.; Takeda, S.; Ito, Y.; Nakajima, T.; Takami, Y.; Takeya, Y.; Yamamoto, K.; Sugimoto, K.; Shimizu, H.; Shimamura, M.; et al. Novel Method for Rapid Assessment of Cognitive Impairment Using High-Performance Eye-Tracking Technology. Scientific Reports 2019, 9, 12932, doi:10.1038/s41598-019-49275-x.
[20] Nie, J.; Qiu, Q.; Phillips, M.; Sun, L.; Yan, F.; Lin, X.; Xiao, S.; Li, X.J.F.i.A.N. Early Diagnosis of Mild Cognitive Impairment Based on Eye Movement Parameters in an Aging Chinese Population. Frontiers in Aging Neuroscience 2020, 12 221, doi:10.3389/fnagi.2020.00221.
Point 6: In particular, it is necessary to explain and support with bibliographic references the statement in line 51 "the weakness of traditional neuropsychological tests is the subjectivity". In fact, the process that allows to build an appropriate and correctly validated neuropsychological test usually leaves no room for subjectivity, as it is always calibrated and standardized tools. It is therefore considered necessary that the meaning of this statement be explained and contextualized with the use of an appropriate bibliography.
Response 6: Thank you for your comment. Neuropsychological tests for MCI screening were designed to capture the slight changes in cognitive decline by individuals’ subjective responses, while neuropsychological tests might be limited in their effectiveness in MCI screening. To avoid readers' misunderstanding, we have rewritten the sentence in 67 to Line 69. Please check this change in the manuscript.
Changes in the manuscript:
Neuropsychological tests might be limited in their effectiveness in MCI screening, while acknowledging that neuropsychological tests inadequate for making a definitive diagnosis.
Point 7:In addition, in lines 59 and 62, two papers are cited without bibliographic references. Please implement these aspects and elaborate on these paragraphs, supporting your statements with appropriate bibliographic references.
Response 7: Thank you for your comment. According to your suggestion, we have extended the manuscript with the corresponding bibliographic references in Line 76 and Line 79
Changes in the manuscript:
For example, Lin et.al. used non-invasive clinical variables and ML classifiers, including Support Vector Machine (SVM), Logistic Regression (LR), and Random Forest (RF), to achieve over 75% classification accuracy to classify subjects who converted to MCI form normal within 4 years [25]. Yim et.al. proposed a ML algorithm to identify cognitive dysfunction based on neuropsychological tests including the Montreal Cognitive Assessment (MoCA). The results showed a good classification performance between cognitive impairment and normal subjects [15].
[25] Lin, M.; Gong, P.; Yang, T.; Ye, J.; Albin, R.L.; Dodge, H.H.J.A.D.; Disorders, A. Big Data Analytical Approaches to the NACC Dataset: Aiding Preclinical Trial Enrichment. Alzheimer Disease & Associated Disorders 2018, 32, 18–27,doi: 10.1097/WAD.0000000000000228.
[15] Yim, D.; Yeo, T.Y.; Park, M.H.J.T.J.o.i.m.r. Mild cognitive impairment, dementia, and cognitive dysfunction screening using machine learning. Journal of International Medical Research 2020, 48,7, doi:10.1177/0300060520936881.
Point 8: Comment 3: in the paragraph Methods the inclusion and exclusion criteria should be better specified and clarified as they are vague and generic for the control group. It is important to explain how the authors detect no memory decline in these subjects. Did they use some neuropsychological or other assessment?
Response 8: Thank you for your comment. Measures and normative means to detect memory decline are long delayed memory and recognition task of Auditory Verbal Learning Test. According to your suggestion, we have revised the inclusion/exclusion criteria of MCI in Line 189 to Line 205 and the inclusion criteria for NC in Line 206.
Changes in the manuscript:
All subjects signed an informed consent before the examinations. This study has been approved by the ethics committee of Long Hua Hospital in Shanghai University of Traditional Chinese Medicine (Ethical number: 2017LCSY345) and conducted in accordance with the principles of the Declaration of Helsinki. In this study, Cohort 1 was used as the training and validation group to train the SVM classifier. Cohort 2 was used as an independent test group to verify the robust of the classification results.
MCI was defined by an actuarial neuropsychological strategy proposed by Jak and Bondi [30], subjects were considered to have MCI if they meet any of the following three criteria and neglect to meet the criteria for dementia. The inclusion criteria for MCI were as follows[31,32]: (1) right-handed, and Mandarin-speaking subjects;(2) a subjective memory complaint; (3) memory impairment relative to age and education-matched healthy elderly individuals confirmed by performance on neuropsychological assessments (below 1.5 standard deviation); (4) intact general cognitive function confirmed by MoCA-B scores >=26; (5) intact activities of daily living; and (6) without dementia confirmed by a physician.
Exclusion criteria of MCI were as follows:(1) other neurological diseases including cerebrovascular disease, brain trauma, Parkinson syndrome, brain tumor, epilepsy; (2) current major psychiatric disease such as severe depression and anxiety; (3) Other neurological conditions that could cause cognitive decline (eg, brain tumors, Parkin-son’s disease, encephalitis, or epilepsy) rather than AD spectrum disorders; (4) systemic diseases that may lead to cognitive decline (thyroid dysfunction, severe anemia, syphilis, or HIV etc.);(5)other conditions such as a history of CO poisoning and general anesthesia; (6) severe visual or hearing impairment. (7) contraindication for MRI.
The inclusion criteria for NC included the following: (1) no subjective or inform-ant-reported memory decline; (2) non-clinical depression (Geriatric Depression Scale scores < 6) ;(3) Normal age-adjusted, gender-adjusted and education-adjusted performance on standardized cognitive tests.
[30] Bondi, M.W.; Edmonds, E.C.; Jak, A.J.; Clark, L.; Delano-Wood, L.; McDonald, C.R.; Nation, D.A.; Libon, D.J.; Au, R.; Galasko, D.R.; et al. Neuropsychological criteria for mild cognitive impairment improves diagnostic precision, biomarker associations, and progression rates. 2014, 42 1, 275-289.
Point 9: Comment 4: In the Discussion section it might be beneficial to discuss the results in more depth in the context of the clinical setting using adequate bibliography.
Response 9: Thank you for your comment. According to your suggestion, we have revised the bibliography in Table 5 and provided a detailed literature review on the state-of-the-arts classification based on EEG model, ET model and NTB model.
Changes in the manuscript:
Cognitive decline remains highly underdiagnosed in the community despite ex-tensive efforts to find novel approaches to detect MCI, to find objective screening methods for cognitive decline could improve early MCI diagnosis. MCI screening in the community has becoming a hot topic nowadays. In sight of their excellent performance in detecting cognitive delice with MCI patients, multimodal detection approaches have been commonly used in computer-aided disease diagnostic fields of community screening. In this study, we proposed a ML model based on EEG, Eye movement and neuropsychological tests for MCI screening in the community level. In contrast of other traditional models, such as EEG based model, ET based model and NTB based model, the classification results of our model outperformed than other traditional models.
So far, a lot of studies have focused on the classification among NC and MCI by using machine learning models for screening in primary care. For instance, Siuly et al. performed a Piecewise Aggregate Approximation (PAA) technique for compressing massive volumes of EEG data for reliable analysis and developed a model based on Extreme Learning Machine (ELM) with permutation entropy (PE) and auto-regressive (AR) model features to achieve the highest MCI classification accuracy (98.8%) [33]; Lagun et al. applied a SVM based machine learning model to arrive the accuracy of 87% to detect MCI by modeling eye movement characteristics such as fixations, saccades, and refixations during the Visual Paired Comparison (VPC) task[34]ï¼›Yim et al. developed a screening model based on gradient boosting(GB) algorithm to identify MCI by neuropsychological test results and arrived the classification accuracy of 93.5%[35];Wang et al. developed a Random Forest(RF)-based model to optimizes the content of cognitive evaluation and achieved an accuracy of 68% in the classification of MCI and NC[36].
According to our work’s benefit in clinical setting, we have supplied the description in Line 314 to 316.
Changes in the manuscript:
3.In terms of clinical setting, we depicted a machine learning framework for automated cognitive assessment data analysis for the precise classification of healthy and mild cognitive impairment individuals. Our work opens the possibility for automated assessment of cognitive function in community screening.
Revision of the bibliography in Table 5:
[33] Siuly, S.; Alçin, Ö.F.; Kabir, E.; Åžengür, A.; Wang, H.; Zhang, Y.; Whittaker, F.J.I.T.o.N.S.; Engineering, R. A New Framework for Automatic Detection of Patients With Mild Cognitive Impairment Using Resting-State EEG Signals. 2020, 28, 1966-1976.
[35] Yim, D.H.; Yeo, T.Y.; Park, M.-H.J.T.J.o.I.M.R. Mild cognitive impairment, dementia, and cognitive dysfunction screening using machine learning. 2020, 48.
[36] Wang, J.; Wang, Z.; Liu, N.; Liu, C.; Mao, C.; Dong, L.-l.; Li, J.; Huang, X.; Lei, D.; Chu, S.; et al. Random Forest Model in the Diagnosis of Dementia Patients with Normal Mini-Mental State Examination Scores. 2022, 12.
Point 10: Comment 5: Authors are asked to pay attention to citations in the text. In the bibliography there are no bibliographic references of the neuropsychological tests that have been used and or cited in the text. Please enter the correct references.
Response 10: Thank you for your comment. According to your suggestion, we had checked and updated the corresponding bibliographic references in the manuscript.
Changes in the bibliography:
Lin et.al. used non-invasive clinical variables and ML classifiers, including Support Vector Machine (SVM), Logistic Regression (LR), and Random Forest (RF), to achieve over 75% classification accuracy to classify subjects who converted to MCI form nor-mal within 4 years [25] in Line 79.
Yim et.al. proposed a ML algorithm to identify cognitive dysfunction based on neuropsychological tests including the Montreal Cognitive Assessment (MoCA). The results showed a good classification performance between cognitive impairment and normal subjects [15] in Line 83.
MCI was defined by an actuarial neuropsychological strategy proposed by Jak and Bondi [28], subjects were considered to have MCI if they meet any of the following three criteria and neglect to meet the criteria for dementia in Line 190.
Siuly et al. performed a Piecewise Aggregate Approximation (PAA) technique for compressing massive volumes of EEG data for reliable analysis and developed a model based on Extreme Learning Machine (ELM) with permutation entropy (PE) and auto-regressive (AR) model features to achieve the highest MCI classification accuracy (98.78%) [33] in Line 286.
Lagun et al. applied a SVM based machine learning model to arrive the accuracy of 87% to detect MCI by modeling eye movement characteristics such as fixations, saccades, and refixations during the Visual Paired Comparison (VPC) task[34] in Line 289.
Yim et al. developed a screening model based on gradient boosting(GB) algorithm to identify MCI by neuropsychological test results and arrived the classification accuracy of 93.5%[35] in Line 291
Wang et al. developed a Random Forest(RF)-based model to optimizes the content of cognitive evaluation and achieved an accuracy of 68% in the classification of MCI and NC[36] in Line 293.

Round 2
Reviewer 1 Report
Dear authors,
Thank you very much for addressing my comments and suggestions. This reviewer considers the manuscript at its present can be accepted.